# The Mechanisms of Action of Hyperbaric Oxygen in Restoring Host Homeostasis during Sepsis

**DOI:** 10.3390/biom13081228

**Published:** 2023-08-07

**Authors:** Julie Vinkel, Bjoern Arenkiel, Ole Hyldegaard

**Affiliations:** 1Department of Anesthesiology, Centre of Head and Orthopedics, Rigshospitalet, University of Copenhagen, 2100 Copenhagen, Denmark; 2Department of Clinical Medicine, University of Copenhagen, 2200 Copenhagen, Denmark

**Keywords:** hyperbaric oxygen treatment, sepsis, host immune response, tolerance to infection, oxygen, hypoxia-inducible factor 1-alpha, nuclear factor kappa-light-chain-enhancer of activated B cells, systemic infectious diseases, inflammation, hypoxia

## Abstract

The perception of sepsis has shifted over time; however, it remains a leading cause of death worldwide. Sepsis is now recognized as an imbalance in host cellular functions triggered by the invading pathogens, both related to immune cells, endothelial function, glucose and oxygen metabolism, tissue repair and restoration. Many of these key mechanisms in sepsis are also targets of hyperbaric oxygen (HBO_2_) treatment. HBO_2_ treatment has been shown to improve survival in clinical studies on patients with necrotizing soft tissue infections as well as experimental sepsis models. High tissue oxygen tension during HBO_2_ treatment may affect oxidative phosphorylation in mitochondria. Oxygen is converted to energy, and, as a natural byproduct, reactive oxygen species are produced. Reactive oxygen species can act as mediators, and both these and the HBO_2_-mediated increase in oxygen supply have the potential to influence the cellular processes involved in sepsis. The pathophysiology of sepsis can be explained comprehensively through resistance and tolerance to infection. We argue that HBO_2_ treatment may protect the host from collateral tissue damage during resistance by reducing neutrophil extracellular traps, inhibiting neutrophil adhesion to vascular endothelium, reducing proinflammatory cytokines, and halting the Warburg effect, while also assisting the host in tolerance to infection by reducing iron-mediated injury and upregulating anti-inflammatory measures. Finally, we show how inflammation and oxygen-sensing pathways are connected on the cellular level in a self-reinforcing and detrimental manner in inflammatory conditions, and with support from a substantial body of studies from the literature, we conclude by demonstrating that HBO_2_ treatment can intervene to maintain homeostasis.

## 1. Introduction

Sepsis and septic shock continue to be a leading cause of death worldwide [1,2,3]. Sepsis was previously thought to be an overwhelming, systemic, proinflammatory response to infection, which was followed by a phase of immunosuppression that was marked by anergy, lymphopenia, and secondary infections [4]. New paradigms suggest that the proinflammatory and immunosuppression phases occurs simultaneously, and the degree to which one response dominates the other depends on the pathogen’s type, virulence, and load as well as the host’s genetics and comorbidities [5]. Two different host defensive tactics can be used to manage an invasive pathogen’s impacts. One strategy, that is also best described in the literature, is referred to as resistance to infection and relies on the host immune system’s capacity to eradicate the pathogen and lower pathogenic load following invasion. When the host immune system fails to return to homoeostasis, it causes irreversible tissue damage and compromises host capability [6,7]. The other defense strategy, known as tolerance to infection, acts independently of microbial load and focuses on tissue damage control, which preserves parenchymal tissue functionality and maintains homeostasis compatible with host survival [8]. While genetically distinct, these two host defense strategies are functionally integrated through mechanisms that are still unknown [9], and both strategies may be equally important targets in sepsis management.

Hyperbaric oxygen treatment (HBO_2_) consists of breathing 100% oxygen under increased ambient pressure, resulting in a rise in the inspiratory partial pressure of oxygen. Hemoglobin is nearly fully saturated under normobaric conditions, therefore the elevated quantity of oxygen is in solution, and the partial pressure of oxygen in the blood stream greatly exceeds the oxygen pressure in the tissue, creating a driving force for oxygen to reach physically obstructed areas where red blood cells cannot pass and can also enable tissue oxygenation even with impaired hemoglobin oxygen carriage [10,11]. Mitochondria use 85–90% of the oxygen in cells for oxygen-dependent oxidative phosphorylation, that sources both the cell’s energy currency ATP (adenosine triphosphate) and the natural byproducts, reactive oxygen species (ROS). ROS overproduction is combated by scavenging antioxidants, these include both enzymatic and non-enzymatic antioxidants like vitamins, uric acid, and carotenes [12]. When present in the right place and at the right concentrations, ROS are also important signaling transduction mediators in cells; however, at high concentrations, they can mediate harmful oxygen toxicity [12]. Mitochondria are found in nearly all cells and are the primary molecular target of HBO_2_ treatment. HBO_2_ treatment has been shown to affect ROS production, particularly via the mitochondria; however, HBO_2_ treatment also affects mitochondrial function, antioxidant levels and activity, effectively lowering ROS levels with repeated treatments [13,14]. Importantly, HBO_2_ treatment is administered in sessions, meaning that after a brief period of hyperoxia (usually 60–120 min), the oxygen level is returned to 21% at normobaric pressure. Such fluctuations in oxygen availability activate redox (reduction and oxidation) sensitive transcription factors and downstream signaling pathways involved in cellular defense mechanisms, antioxidant activities, metabolism, cell survival, inflammation, and immune responses [13,15,16,17]. 

The redox cellular balance is critical for normal cell function, and severe oxidative stress occurs in sepsis, which must be counterbalanced by the host’s endogenous antioxidant defenses to maintain homeostasis [18]. This opens the door to using HBO_2_ treatment as an adjunctive measure to counteract the negative effects of sepsis. Although HBO_2_ treatment is not currently recommended for sepsis, experimental studies have shown that it improves survival [11,19,20]. Furthermore, clinical studies evaluating the effects of HBO_2_ treatment in patients with sepsis caused by necrotizing soft tissue infections (NSTIs) have revealed a link between improved survival and levels of key clinical markers. [21,22,23,24]. In this perspective, it is noteworthy to emphasize that the connection between HBO_2_ and better survival in NSTIs is stronger in the most critically ill with septic shock [25,26]. However, because there are no clinical randomized studies on the effects of HBO_2_ in a cohort of patients with severe infections, the intervention’s potential effects and processes must be investigated before HBO_2_ treatment for sepsis may be considered.

In the sections that follow, we will discuss the potential mechanisms of action of HBO_2_ treatment to maintain host homeostasis, in terms of both resistance and tolerance to infection.

## 2. Resistance to Infection

The term “infection resistance” refers to all host processes initiated to reduce pathogen load during an infection. Invading microorganisms exhibit pathogen-associated molecular patterns, which are recognized as alien by host PRRs (pattern recognition receptors), and local infection damages host cells, resulting in the release of cellular constituents, which are also recognized as alien by the PRRs. Thus, PRRs on circulating immune cells are activated, which causes transcription factors to enter the cell nucleus and either suppress or increase transcription of genes that coordinate inflammatory responses [5,27]. During a controlled infection, circulating immune cells are drawn to the infection site to phagocytose and remove damaged host cells and pathogenic microorganisms, disrupting the preceding chain of events and preserving host homeostasis [28]. 

### 2.1. Neutrophil Mediated Responses in Sepsis

Neutrophils play a central role in the resistance to infection by removing invading pathogens through phagocytosis [29]. Other important functional responses that neutrophils use in host defense include cell migration, degranulation, and oxidative bursts. Oxidative bursts are a potent antibacterial tool against bacterial and fungal infections, and they require the fast synthesis of ROS by the NADPH (nikotinamid-adenin-dinucleotidfosfat) oxidase complex [30]. ROS can be released extracellularly into the environment surrounding the site of infection or intracellularly in the phagolysosome following phagocytosis of bacteria. ROS improves neutrophil antimicrobial response by stimulating the production of inflammatory cytokines, activating granule release, and generating neutrophil extracellular traps (NETs) [30]. NETs can entrap pathogens and thereby contribute to pathogen elimination. However, uncontrolled NETs’ release can also contribute to collateral tissue damage and thrombosis by serving as a scaffold for the entrapment and aggregation of platelets and erythrocytes [28]. As a result, while oxidative bursts contribute to host defense as described, they can also cause collateral damage to host tissues [31]. Patients with sepsis have elevated NET levels in their blood, which are linked to organ dysfunction [32]. HBO_2_ treatment has a complex effect on the host’s oxidative status, which is likely to be dependent on both dose and timing, as well as tissue oxygenation at the time of application. HBO_2_ has been shown in cells from healthy volunteers to reduce ROS production by neutrophils after two and three HBO_2_ treatment sessions [33]. The same study found no change in neutrophil phagocytic activity, circulating cytokines, or systemic oxidative stress, as indicated by plasma malondialdehyde concentrations [33]. Similar results were found in a study evaluating the effects of HBO_2_ treatment ex vivo on the activity of neutrophils harvested from severely injured patients. HBO_2_ treatment significantly decreased neutrophil ROS production, which was also linked to a decrease in the release of neutrophil extracellular traps (NETs) in control cells and to a lesser extent in cells from injured individuals [32]. HBO_2_ did not influence neutrophil chemotaxis or apoptosis [32].

### 2.2. Endothelial Dysfunction in Sepsis

Sepsis is associated with severe endothelial cell dysfunction. Endothelial cell lining integrity is determined largely by the endothelial cytoskeleton and the glycocalyx. Glycocalyx shedding causes barrier dysfunction, is linked to edema formation, and is a major contributor to sepsis-induced organ failure. [34]. Cytokines and reactive species induce glycocalyx shedding in inflammation and sepsis, exposing adhesion molecules and initiating leukocyte adhesion, which leads to transmigration to tissues. Reactive oxygen species (ROS) such as hydrogen peroxide, hydroxyl anions, and superoxide are thought to be the primary inducers of glycocalyx shedding, whereas integrins such as 2 integrins bind to ICAM-1 (intercellular adhesion molecule-1) and mediate adhesion and transcellular trafficking of leukocytes to parenchymal cells [35,36]. Once they have entered the tissues, leukocytes can release inflammatory mediators and reactive molecules to destroy pathogens, but at the same time potentially cause tissue damage [34]. Given the central role of ROS and reactive nitrogen species, HBO_2_ treatment may modulate the cascade of events that results in organ failure caused by endothelial cell dysfunction. Indeed, HBO_2_ treatment has been demonstrated to mediate inhibition of neutrophil adhesion to vascular endothelium [37]. The effect has been shown to be specific to β2-integrin class [35,38], and it occurs because hyperoxia increases the activity of NOS (nitric oxide synthases) and MPO (myeloperoxidase) in neutrophils, resulting in the release of NO (nitric oxide)-derived oxidants, that mediate excessive S-nitrosylation of β-actin, a cytoskeleton actin in neutrophils required for β2-integrin clustering. It is a localized process occurring only within neutrophils, probably because of a scarcity of myeloperoxidase [39]. On the endothelial side, a prospective study of patients with sepsis caused by necrotizing soft tissue infections showed that HBO_2_ treatment increased soluble ICAM-1 (sICAM-1) and the effect was more pronounced in patients with septic shock. Low baseline sICAM-1 was an independent risk factor of 90-day mortality and was associated with severity of disease [22]. HBO_2_ treatment might modulate endothelial shedding of ICAM-1 and reduce the inflammation on the endothelium. This is in line with an in vivo study demonstrating that HBO_2_ treatment reduces ICAM-1 on vascular endothelium under infectious conditions [40].

### 2.3. The Cytokine Mediated Inflammatory Response in Sepsis

Substantial in vitro and in vivo evidence suggests that NF-κB (Nuclear Factor kappa-B) plays a critical role in sepsis [18]. NF-κB is a transcription factor family that is involved in a variety of biological responses that underpin the phenotypic outcomes of inflammation, immune response modulation, cell growth, proliferation, apoptosis, and aspects of differentiation and development. NF-κB signaling depends on the mobilization of both homo- and hetero-dimer complexes of these family members. In an inactive state these proteins are typically associated with inhibitory-κB (IκB) proteins [41]. The translocation of NF-κB into the nucleus and the subsequent activation of genes, including those coding for cytokines is crucial for the induction of inflammation. Pro-inflammatory cytokines that are believed to be involved in excessive inflammation in sepsis are TNFα (tumor necrosis factor-alpha), IL-1β (interleukin-1β), IL-12 (interleukin-12), and IL-18 (interleukin-18) [28]. In addition to initiating the immuno-inflammatory response, cytokines also coordinate and modulate the nature, amplitude, and duration of the response [18]. Increased and/or prolonged activation of NF-κB result in the overexpression of mediator proteins and may account for some of the deleterious effects seen in sepsis [18]. A recent systematic review with meta-analysis by Gharamti et al. revealed that the mean TNFα concentration in sepsis was approximately 10-fold higher than the mean concentration in healthy individuals, and that TNFα was associated with sepsis mortality but not sepsis severity [42]. A prospective cohort study not included in the meta-analysis by Gharamti et al. including 242 patients with sepsis due to necrotizing soft tissue infection found significantly higher baseline values of TNF-α, IL-1β, IL-6 (interleukin-6), and G-CSF (granulocyte colony-stimulating factor) in patients presenting with shock, compared to non-shock patients [23]. Likewise, a separate group has found that concentrations of the pro-inflammatory cytokines IL-1β, IL-6, IL-7 (interleukin-7), IL-8 (interleukin-8), IL-13 (interleukin-13), interferon-γ, monocyte chemoattractant protein-1, and TNF-α were significantly higher in septic shock patients than in those with severe sepsis, and the expression of pro-inflammatory cytokines was associated with organ failure and mortality [43]. NF-κB is a redox-sensitive transcription factor, and exposure to oxidants such as hydrogen peroxide causes nuclear translocation of NF-κB in certain cells [44]. Interestingly, HBO_2_ treatment has been shown to decrease the expression of NF-κB at protein level in both an LPS (liposaccharide) model of sepsis, neuroinflammation, healthy cells, and cancer cells [45,46,47], preventing the production of inflammatory cytokines and pulling in an anti-inflammatory direction under ongoing stress [46,48]. HBO_2_ treatment has been shown to reduce proinflammatory cytokines such as IL-1, IL-6, and TNF-α both in vitro and in vivo [49], and HBO_2_ treatment was associated with a decrease in IL-6 and G-CSF in plasma from Group A-Streptococcus NSTI sepsis patients [23]. However, some studies have not found any changes in cytokine levels after HBO_2_ treatment [33]. A systematic review of 58 papers on inflammatory markers in human tissue in response to HBO_2_ treatment found an inhibiting effect on NF-κB, IL-1β, IL-6, and IL-8, as well as an anti-inflammatory state in general [48]. 

### 2.4. Switching Immune Cell Metabolism toward Glycolysis in Sepsis—The Warburg Effect

Mitochondria provide the energy required for normal cellular activity, including the ability to respond to any pathophysiological stress, in the form of ATP. ATP is normally produced in the cytosol by glycolysis or, to a much greater extent, oxidative phosphorylation. Otto Warburg discovered that cancer cells produced energy via glycolysis rather than oxidative phosphorylation [50]. The Warburg Effect is increasingly recognized as an essential regulator of innate and adaptive immunity and may be as much a hallmark of sepsis as it is of cancer [51,52,53,54]. The Warburg effect may help to regulate innate immune functions in activated immune cells like macrophages, dendritic cells, and T cells by providing a readily available source of energy for phagocytosis, oxidative burst, and biosynthetic precursors to divide and produce cytokines, and thus acts as a way to regulate host resistance to infection [55]. The transcription factor hypoxia inducible factor-1 (HIF-1) is thought to be responsible for the switch toward glycolysis in sepsis via the epigenetic upregulation of HIF-1 target genes expressing mTOR. In activated lymphocytes, the protein kinase mTOR (mammalian target of rapamycin) acts as a sensor of the metabolic environment and as a master regulator of glucose metabolism [52]. It is possible that hypoxia contributes to the Warburg effect because it occurs as part of the pro-inflammatory response in sepsis and is known to stimulate glycolysis in inflammation and cancer. However, glycolysis continues in both pathologies despite adequate oxygen delivery to the tissues. It is assumed that the biochemical events that allow glycolysis to occur even in the presence of oxygen are similar in sepsis and cancer. One proposed mechanism is that pyruvate is kept in the cytoplasm, where it can be reduced to lactate even in the presence of oxygen, preventing it from being used in the tricarboxylic acid cycle [53]. Excessive pro-inflammatory glycolytic drive may have a negative impact on host cell homeostasis. Lactate is a byproduct of glycolysis, and higher levels of lactate are associated with a worse outcome in sepsis disorders [56,57]. Lactate elevation is commonly thought to occur as a result of poor microcirculatory perfusion, which causes a switch to anaerobic respiration; however, recent studies indicate that lactate is more than just a marker of circulatory abnormalities, and most likely represents a fundamental shift in metabolism to a more proinflammatory glycolysis [53,58,59]. In line with this, an experimental study found that the PKM2 (pyruvate kinase M2)-mediated Warburg effect contributes to sepsis mortality, presumably because PKM2 interacts as a coactivator with HIF-1 and activates the transcription of glycolysis-related genes, resulting in excessive lactate production [54]. HBO_2_ treatment can stimulate changes in cellular energy metabolism thereby halting the Warburg effect [60]. Because sepsis and cancer are thought to have similar biochemical events in terms of the Warburg effect, studies on HBO_2_ treatments’ effect on the Warburg effect in carcinogenic illnesses may be generalized to sepsis situations. HIF-1 downregulation was required for HBO_2_ treatment to suppress the Warburg effect in hypoxic cancer cells, according to in vitro research on non-small cell lung cancer cell lines [61]. Another study found that HBO_2_ reduced oxidative phosphorylation during both the initial response phase and the recovery phase of energy production. HBO_2_ also downregulated ribosomal protein S6 kinase, a target of the mTOR pathway [60]. Furthermore, a few studies have shown that a ketogenic diet combined with HBO_2_ treatment can limit tumor growth in experimental models of metastatic cancer by blocking the Warburg effect [61,62,63]. 

In short, the host effector mechanisms discussed above that act to increase resistance to infection and fight the invading pathogen may also reduce tolerance to infection by causing self-harm [7], and the HBO_2_ treatment acts to counterbalance collateral damage during resistance and increasing tolerance, as shown in Figure 1.

## 3. Tolerance to Infection

The term “disease tolerance” refers to a biological phenomenon originally identified in host–microbe interactions in plants. The concept involves all host biological mechanisms that limit the impact of infection on host integrity and fitness without interfering with the pathogen burden, thereby preserving functionalities and preventing tissue damage and organ dysfunction [8,9]. Organ dysfunction in sepsis is an important predictor of patient outcome, and tissue hypoxia, mitochondrial dysfunction, and apoptosis are all thought to be important mediators of sepsis-induced organ dysfunction. Tissue ischemia can occur because of a systemic or local mismatch between oxygen delivery and tissue demand. On a cellular level, mitochondrial dysfunction can also result in a failure of tissue oxygen extraction despite adequate oxygen delivery, a condition known as cytopathic hypoxia [64]. 

### 3.1. Iron Metabolism in the Pathogenesis of Sepsis

In severe bacterial infections, invading bacteria are frequently transferred from the local infectious site to the bloodstream, where they can cause hemolysis, which results in the oxidation of extracellular hemoglobin and the accumulation of labile heme in plasma. [6,8]. In the presence of a pro-inflammatory agonist, free heme can cause irreversible tissue damage and organ failure through a cytotoxic effect based on its pro-oxidant activity and the production of free radicals via the Fenton reaction [6]. To increase their tolerance for infection, the host has defenses against the cytotoxic effects of heme. HO-1 (Heme oxygenase-1) catabolizes heme and breaks it down into biliverdin, iron (Fe), and carbon monoxide [65]. In a public transcriptome dataset of patients with sepsis, HO-1 expression was found to be higher in non-survivors than in survivors, and the authors interpreted it as an expression of higher disease severity of non-survivor patients [66]. This is supported by several studies in mice, with one finding that when exposed to an equal pathogen load, mice that can stimulate the expression of HO-1 in response to polymicrobial infection have a higher chance of surviving than mice with HO-1 deficiency [6]. Another study revealed that a plasmodium-infected host’s ability to survive solely depends on its capacity to counteract the cytotoxic effects of free heme by expressing HO-1 [67]. The protective effect of HO-1-mediated heme catalysis, on the other hand, has a payoff in the generation of labile iron, which can catalyze the production of reactive oxygen species [68]. Ferritin and its component, which has ferroxidase activity, counteract this, also increasing the host’s tolerance to infection [8]. The expression of HO-1 is dependent on the cell type, cellular microenvironment, intensity and duration of stimuli exposure, and is regulated by a panel of redox-sensitive transcription factors, including HIF-1α and NF-κB [69]. In numerous studies and circumstances, HBO_2_ treatment has been linked to a protective impact of enhanced HO-1 activity [70,71,72,73,74,75]. Most pertinent to this review is that levels of HO-1 have been shown to increase in a subgroup of NSTI patients with septic shock in response to HBO_2_ treatment [24]. Other authors have likewise demonstrated an effect of HBO_2_ treatment on HO-1 with a 30-fold increase in lymphocytes of healthy volunteers 24 h after HBO_2_ treatment [73]. The beneficial effects of HO-1 stimulation in sepsis have been demonstrated in a lipopolysaccharide-induced model, demonstrating that HO-1 induction reduces acute lung injury [70]. 

### 3.2. The Anti-Inflammatory Response in Sepsis

According to recent studies on the pathophysiologic mechanisms underlying sepsis, specific anti-inflammatory cytokines balance out the strong proinflammatory response to restore immunological balance and increase host resistance to infection [27]. Proinflammatory cytokines, anti-inflammatory cytokines, and soluble inhibitors of proinflammatory cytokines make up the cytokine network, and a tightly controlled balance between these three groups of cytokines is essential for preventing excessive, tissue-damaging inflammation during the thorough removal of invading pathogens [76,77]. IL-10 (interleukin-10), IL-4 (interleukin-4), and TGF-*β* (tumor growth factor beta) are the three best described anti-inflammatory regulators operating in sepsis. IL-10 is produced by many types of immune cells, such as monocytes, macrophages, B- and T lymphocytes, and natural killer cells. Studies conducted in vitro have demonstrated that IL-10 inhibits immune cells’ production of proinflammatory mediators such as TNFα, IL-1, IL-6, and IFN-*γ* (interferon-gamma) [78]. This is supported by in vivo studies that demonstrate IL-10 has a protective effect in mice with lethal endotoxemia [79,80]. Several studies have found that IL-10, along with a panel of pro-inflammatory cytokines, is expressed in severe sepsis, indicating a role in limiting excessive inflammation [23,43]. IL-4 is a critical regulator of T lymphocyte differentiation, promoting Th2 cell differentiation while also being the primary cytokine produced by Th2 lymphocytes. As a result, more IL-4 and other anti-inflammatory cytokines are released, while monocyte-derived proinflammatory cytokines are suppressed [81]. Every cell in the body, including epithelial, endothelial, hematopoietic, neuronal, and connective-tissue cells, produces TGF-β (transforming growth factor-beta) and has receptors for it. TGF-β1 levels in plasma have been found to be significantly higher in septic patients than in healthy donors, and platelets may be an important source of this cytokine during sepsis, the levels of TGF-β1 were, however, not correlated with outcome [82]. There have been few investigations on the influence of HBO_2_ medication on the production of anti-inflammatory cytokines in sepsis, with inconsistent results. One clinical study found no effect of HBO_2_ treatment on IL-10 in septic patients after one or three treatment sessions [23]. In a cecal ligation and puncture-induced sepsis model macrophages isolated from HBO_2_-treated mice demonstrated enhanced IL-10 secretion as compared with controls, and IL-10 deficiency mice were not protected from sepsis mortality by IL-10 expression [83]. A protective effect of IL-10 elevation in response to HBO_2_ treatment has also been demonstrated in another model [84]. Animal studies have found that HBO_2_ treatment decreases the level of TGF-β messengers and proteins in several conditions [85,86,87,88], while IL-4 is upregulated in response to HBO_2_ treatment [89,90,91]. This upregulation of IL-4 was confirmed in a systematic review on cytokine responses to HBO_2_ in human tissue, whereas the authors found no effect on IL-10 and a possible reducing effect on TGF-β [48].

### 3.3. Hypoxia in the Pathogenesis of Sepsis

In disease pathology, hypoxia and inflammation go hand in hand. During an infection, tissue hypoxia is caused by vascular dysfunction and increased oxygen consumption due to increased metabolic activity of immune cells. Hypoxia acts as a stressor in cells, activating NF-κB, which then upregulates hypoxia-inducible factors (HIFs) and the cellular hypoxia response in a positive feedback loop [92,93]. This might culminate in reinforcing pathophysiological effects in which both hypoxia and inflammation contribute to disease [94]. 

Hypoxia inducible factors (HIFs) are transcription factors that act in all cells and are known to be important regulators of oxygen homeostasis and in pathological conditions [95,96]. Under normal conditions, HIF proteins are constantly produced by transcription and translation, but HIF subunits and messenger RNA have a very short half-life of about 5 min [95]. HIFs are regulated by two oxygen-dependent inhibitors: PHD (prolyl hydroxylase domain), which contains oxygen-sensing hydroxylases that cause its degradation, and FIH (factor-inhibiting HIF), which controls the transcription factor HIF [14,94,97]. In hypoxic conditions, HIF-1α is stabilized, translocated to the nucleus, dimerizes with the constitutively expressed HIF-β, and binds to a hypoxia response element in the DNA (Figure 2). HIFs directly or indirectly regulate more than 100 genes, the products of which restore blood supply, nutrients, and energy production, thereby maintaining tissue integrity and homeostasis, making HIF-1 a key player in host tolerance to infection [66,95,97]. 

Experimental studies have shown that HIF-1 may be a critical determinant of the sepsis phenotype due to its association with the production of pro-inflammatory cytokines, which results in the clinical manifestation of sepsis symptoms such as tachycardia, hypotension, and hypothermia [98,99]. A study that used transcriptomic profiling to identify sepsis endotypes in intensive care unit patients discovered two distinct endotypes, one of which was associated with higher mortality and features of a maladaptive immune response. The authors identified regulatory genetic variants in this group that account for the upregulated expression of the genes HIF1A and EPAS1 (endothelial PAS domain protein 1), indicating that HIF genetic variants may be deterministic of host sepsis response [100]. In a study of patients with shock, including septic shock, HIF-1α mRNA levels were significantly higher than in healthy volunteers. HIF-1α expression did not differ depending on the type of shock, and there was no correlation between HIF-1 expression and clinical markers of disease severity. However, HIF-1α expression was significantly higher in patients who received more than two liters of fluid expansion, with no correlation to vasopressor dose. The authors hypothesized that the effect on HIF-1α was caused by large fluid resuscitation or vasodilation, which could lead to tissue hypoxia [67]. Using public transcriptome datasets from septic patients’ mononuclear cells, another study discovered that HIF-1α mRNA expression and target genes were higher in non-survivors than survivors on day 7 after admission to the intensive care unit [66]. In contrast, an earlier study found that leukocyte HIF-1α mRNA and HIF-1α protein positive cells were lower in adult patients with severe sepsis compared to healthy volunteers 24 h after diagnosis, and that this was inversely correlated with illness severity, and the authors speculated that the patient immune defenses may have been imbalanced toward immunosuppression at the time of sampling [101]. Another possible explanation is that the patients were not in a state of vascular dysfunction and volume overload 48 h after being diagnosed with sepsis.

It has been proposed that changes in oxygen availability, rather than constant hypoxia or hyperoxia, have a more dominant effect on HIF-1α expression, making HBO_2_ treatment a potent regulator of HIF expression [102,103,104]. The effect of HBO_2_ treatment on HIF is multi-faceted and may occur both at the transcriptional and post-transcriptional level mediated by the actions of oxygen and reactive oxygen species. The treatment-mediated oxygen supply will result in the O_2_-sensitive hydroxylation of HIF-1α proteins and its inactivation via the actions of PHD and FIH. The exact mechanism by which oxygen sensing regulates the NF-κB pathway is still a matter of debate [105,106,107]. Both PHDs and FIH have been identified as playing a role in the control of NF-κB activity in hypoxia, either by directly impacting IKKβ (inhibitor of nuclear factor kappa B kinase subunit beta) activity, or through hydroxylase activity, ultimately leading to hindered tyrosine phosphorylation of IκBα (NF-kappa-B inhibitor alpha), and the continuous binding of this inhibitory protein (Figure 2) [106]. Increased levels of oxygen availability in response to HBO_2_ treatment will also enhance the production of both mitochondrial ROS and ROS scavengers. It is hypothesized that after one or a few hyperoxic exposures, the ratio between ROS and their scavengers will be high, and more ROS will be present [14], while some studies indicate a lowering effect of HBO_2_ treatment on ROS levels in stressed conditions [32,108]. The influence of mitochondrial ROS in cellular oxygen sensing has been widely debated in the last few decades [109,110,111], and may not perpetuate across the conditions of hypoxia to hyperoxia. On the transcriptional level, increased ROS has been demonstrated to increase gene transcription of HIF-1α mRNA in normoxia via binding of NF-κB to a site in the HIF-1α promoter [92,112]. ROS are also demonstrated to regulate HIF-1α at protein level through both PDH-mediated stabilization and posttranslational modifications with NO and sulfur oxide as mediators [110]. However, there is no clear consensus on the exact mechanism(s) by which ROS regulates HIF-1α under hypoxia or normoxia, and, most lately, it is suggested that ROS are not directly involved in HIF-1α expression under hypoxia [110,111].

If this is the case, then when HBO_2_ treatment is administered in hypoxic inflammatory conditions, the regulation of HIF-1α will be primarily driven by increased oxygen delivery, which causes HIF-1α to be hydroxylated and degraded via the actions of PHD, thereby decreasing the expression of HIF-1α proteins [95]. This is in line with the previous observation that HBO_2_ administrated to non-inflamed well-oxygenated tissue causes an increase in HIF-1α protein levels. However, when given to inflamed and ischemic tissue that overexpresses HIF-1α, the overexpressed HIF-1α is reduced. [14]. The oxygen-mediated activation of PHD and HIF will leave NF-κB bound to its inhibitory IκBα preventing its target genes to be transcribed [106]. As a result, in septic conditions, HBO_2_ treatment may break the positive feedback loop mechanism by which hypoxia induces inflammation and inflammation induces hypoxia, thereby preventing the damage caused by this cycle (Figure 2) [94].

To assess the literature on the effects of HBO_2_ treatment on HIF-1 and NF-κB signaling, we conducted a systematic search of the PubMed and Embase databases from their inception to July 2023. The retrieval key words were “HIF” AND “hyperbaric” or “NF-κB” AND “hyperbaric” in English. We excluded studies in which HBO_2_ treatment was only tested in conjunction with another treatment, as well as studies in which full-text papers were not available in English. In studies where HBO_2_ treatment was applied as a pre-conditioning treatment, we included only data comparing healthy controls with healthy controls receiving HBO_2_. Two independent researchers assessed each paper, and any disagreements were discussed with a third researcher. Following a thorough review of the whole text, 54 and 20 papers on HIF-1α and NF-κB, respectively, were included (see Appendix A. The studies indicated that the effect of HBO_2_ is less pronounced on mRNA HIF expression than on protein expression, with four [113,114,115,116] of eleven studies [113,114,115,116,117,118,119,120,121,122,123] showing no effect on mRNA expression and only one out of fifty studies showing no effect on protein expression [17]. HIF-1α expression increased when HBO_2_ is applied to a normoxic environment [114,117,118,124,125,126,127,128], except in one study [17]. Conversely, when HBO_2_ treatment is administered in hypoxic conditions with a derivational elevated HIF-1α, HBO_2_ treatment decreased the disease-mediated high levels of HIF-1α [61,119,129,130,131,132,133,134,135,136,137,138,139]. The evaluation of studies performed in conditions of relative hypoxia secondary to inflammation indicates that the effect of HBO_2_ on HIF-1α protein expression is dependent on the number of HBO_2_ sessions administered, with HBO_2_ having an increasing effect when administered in up to five sessions [140,141,142,143], and treatment with more than six sessions having a decreasing effect under these conditions [47,120,144,145,146,147,148,149,150,151,152,153,154,155,156]. However, six study results were not supportive of this trend [117,140,141,157,158,159]. In general, downstream DNA signaling increased or decreased in direct proportion to changes in the HIF-1 protein [61,114,125,142], except in one study, where an increase in HIF-1 protein increased HIF-1 DNA binding activity but not HIF target gene transcriptional activity (Part a in Table 1) [141].

Studies looking at the impact of HBO_2_ on the expression on NF-κB similarly found an increasing effect in a normoxic environment [17,117,160] and a decreasing effect in a hypoxic environment [161,162], regardless of the stage of inflammation. In contrast to HIF-1α expression, studies evaluating the effect on NF-κB signaling in conditions of relative hypoxia secondary to inflammatory conditions mostly found a decreasing effect regardless of the inflammatory state or number of HBO_2_ sessions [46,47,153,154,163,164,165,166,167,168,169,170], with the exception of three studies where 10–30 sessions of HBO_2_ were applied to chronic inflammation, which found an increasing effect [117] or no effect (Part b in Table 1) [149,158].

**Table 1 biomolecules-13-01228-t001:** The effect of hyperbaric oxygen treatment on (a) HIF-1α expression and (b) NF-κB expression varies according to the inflammatory stage (rows), number of hyperbaric sessions (columns), and oxygen environment before intervention (blue = normoxic, yellow = relative hypoxic, red = hypoxic). A rising arrow indicates an increasing effect, a downward arrow indicates a decreasing effect, and a right arrow indicates no effect. Studies that appear twice or more in the matrix report results on multiple conditions. The studies indicated with an * are ex vivo studies in which live isolated cells were treated with HBO_2_, whilst the remaining studies are in vivo investigations of the effects of HBO_2_ intervention.

		Number of Hyperbaric Sessions						
		1–5 Sessions			6–15 Sessions		16–40 Sessions	
		mRNA	Protein	DNA Signal	mRNA	Protein	mRNA	Protein
Inflammatory stage	None	[113]→*, [117]↑*, [118]↑*, [114]→	[117]↑*, [118]↑*, [124]↑, [125]↑, [114]↑, [17]→, [126]↑	[125]↑, [114]↑		[124]↑, [126]↑		
		[115]→*						
	Acute		[129]↓, [130]↓, [61]↓*, [131]↓, [132]↓, [133]↓, [134]↓, [135]↓,	[61]↓,		[136]↓, [137]↓, [139]↓,		
		[116]→	[171]↓, [172]↓, [140]↑, [141]↑	[141]→*	[120]↓, [121]↑	[120]↓, [144]↓, [145]↓, [146]↓, [147]↓, [156]↓, [148]↓*, [157]↑		
	Chronic		[127]↑*,			[128]↑,		
		[119]↓*	[119]↓*, [138]↓,		[122]↓*,	[122]↓*,		
			[142]↑, [143]↑	[142]↑	[123]↑	[149]↓, [151]↓, [150]↓, [152]↓, [153]↓	[117]↑	[117]↑, [154]↓, [47]↓, [155]↓, [158]↑, [159]↑
(a)								
Inflammatory stage	None	[117]↑*	[160]↑, [17]↑, [117]↑*					
	Acute		[161]↓,			[162]↓,		
			[163]↓, [164]↓, [165]↓, [166]↓, [46]↓, [167]↓	[168]↓	[163]↓, [169]↓	[163]↓, [166]↓, [169]↓		[166]↓
	Chronic					[153]↓, [170]↓, [149]→	[117]↑	[47]↓, [154]↓, [158]→, [117]↑
(b)								

Overall, this supports the presented theory that HBO_2_ treatment decreases HIF-1α and NF-κB signaling during hypoxia, whereas HBO_2_ treatment increases HIF-1α and NF-κB levels during normoxic homeostatic conditions. However, when administering HBO_2_ to conditions of relative hypoxia caused by inflammation, the effect on HIF-1α protein expression may depend on the number of sequential HBO_2_ treatment sessions, whereas the effect on NF-κB expression is primarily decreasing, which may reflect that NF-κB signaling is influenced through multiple pathways during inflammation.

## 4. Discussion

Collectively, this review comprehensively presents some of the lines of defense used by the host to maintain homeostasis in its own ranks while fighting invading pathogens, and we propose how HBO_2_ treatment may assist the host in this complex task by dampening proinflammatory responses and ameliorating tissue damage. Pathways related to oxygen- and inflammatory signaling are interlinked on the cellular signaling level [38,84], and reinforcements in this cause-and-effect system may be hampered by intervening with intermittent controlled HBO_2_ treatment. In our discussion of HBO_2_ treatment effects in connection with host homeostasis, we included studies on a range of conditions, and we recognize that this approach grossly simplifies complex pathophysiological processes in specific diseases. Aside from host homeostasis, HBO_2_ treatment has been shown to have other beneficial effects in combating invading pathogens, such as a synergistic effect with certain antimicrobial agents [173,174], a bactericidal effect on antibiotic-resistant isolates [175], and possibly a reduction in the production of toxins by some bacteria during infection [176]. However, this review provides an overview of some explanatory factors for how and why HBO_2_ treatment has been associated with decreased mortality in systemic infectious diseases, as well as pathways that could be important in the pursuit of HBO_2_ mechanisms of action in host during sepsis.

## 5. Future Directions

The complexity and heterogeneity of sepsis are increasingly recognized by the scientific community, and the nature of this disease may reflect the inconsistencies in biomarker findings that are up- and downregulated during the disease course. As conclusively reported in a recent systematic review and meta-analysis on cytokine levels in sepsis, elevated cytokine levels may have less explanatory power in sepsis pathogenesis than previously thought [32], and mechanisms that regulate other aspects of host tolerance and resistance to infection may be equally important. When an intervention like HBO_2_ treatment, which presumably interacts with biological pathways at multiple levels, is added to this complexity, it is no longer practical to look for associations with a panel of predetermined biomarkers. To gain a better understanding of the mechanisms of action of HBO_2_ treatment in sepsis and severe infections, a systems medicine approach is required.

## Figures and Tables

**Figure 1 biomolecules-13-01228-f001:**
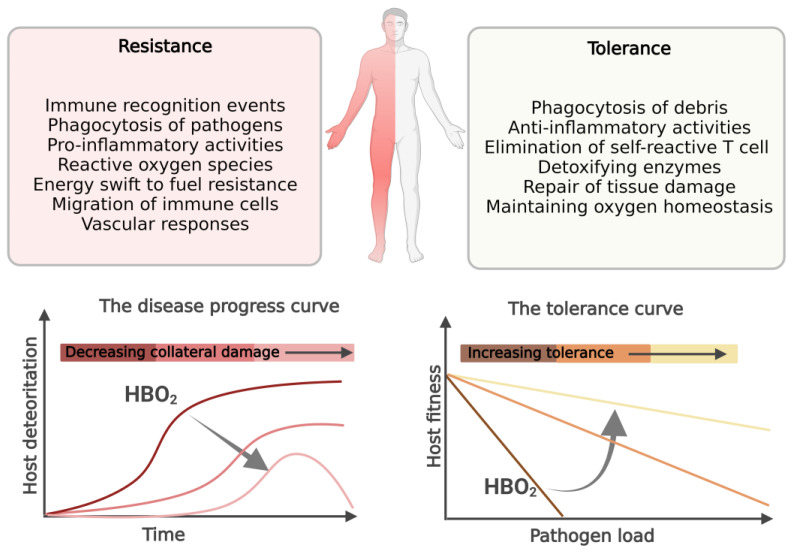
The division of host defense mechanisms in infection resistance and tolerance. Host resistance effector mechanisms that restrict and contain invading pathogens may be self-harming. Disease tolerance is defined as the host’s ability to limit damage and maintain health in the face of increasing pathogen burden. The tolerance curve: in this linear relationship, hosts with steep negative slopes lose health as pathogen loads increase, whereas hosts with shallow slopes maintain relatively higher levels of health even as pathogen burden increases. We propose that HBO_2_ treatment can help the host immune system by increasing tolerance and preventing self-damage during resistance.

**Figure 2 biomolecules-13-01228-f002:**
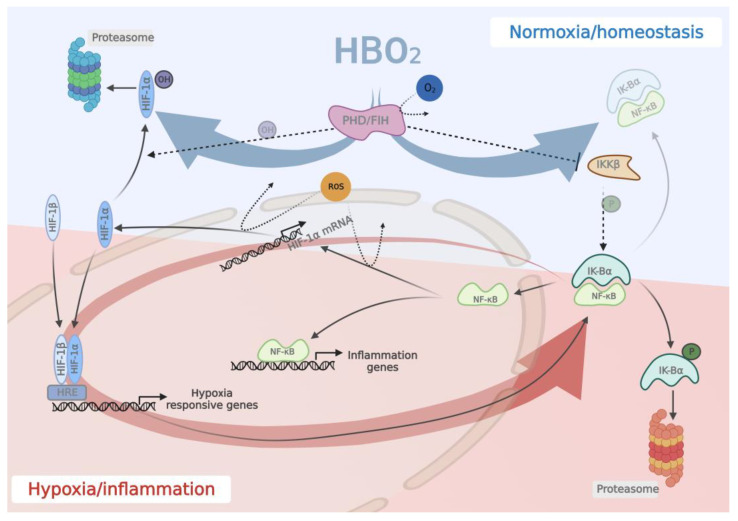
Schematic illustration of the link between the inflammatory NF-κB and hypoxic HIF pathways. During inflammation, a self-perpetuating cycle occurs, in which the host immune balance shifts to a proinflammatory hypoxic state, as seen in sepsis. HIF-1α stimulates the activation of NF-κB genes, resulting in the transcription of inflammation-related genes. Before and during inflammation, NF-κB is a critical transcriptional activator of HIF-1α. High levels of HIF-1α and its constitutively expressed *β*-subunit are translocated to the nucleus in the absence of oxygen, where they bind as heterodimers to HRE, causing transcription of hypoxia responsive genes, including those coding for NF-κB. A loop by which hypoxia and inflammation reinforce one another is established (red arrow). This maladaptive response is interrupted by intervention with HBO_2_. The high levels of dissolved O_2_ during HBO_2_ treatment will activate cellular O_2_-sensing causing hydroxylation and hence proteasomal degradation of HIF-1α and inhibition of IKKβ-mediated tyrosine phosphorylation of IκBα, which keeps it bound to NF-κB in plasma, thereby preventing NF-κB pathway activation (blue arrows). HIF-1α = Hypoxia-Inducing Factor 1-alpha; HIF-1ß = Hypoxia-Inducing Factor 1-beta; HRE = hypoxia response promotor; NF-κB = Nuclear factor kappa-light-chain-enhancer of activated B cells; IκBα = nuclear factor of kappa light polypeptide gene-enhancer in B-cells inhibitor, alpha; PHD = prolyl hydroxylase domain; FIH = Factor Inhibiting HIF-1α; IK-Bα = IkappaB kinase, alpha; OH = Hydroxylation; P = phosphorylation.

## Data Availability

Not applicable.

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
