# Peer review of "The Mechanisms of Action of Hyperbaric Oxygen in Restoring Host Homeostasis during Sepsis"

_biomolecules, 2023, doi:10.3390/biom13081228_

Round 1
Reviewer 1 Report
This is an excellent review of the topic. Good manuscript, with nice pictures, and clear explanations of difficult aspects. Congratulations.
I have noticed that two actions of HBO2 are somehow missing or - at least - not emphasized enough, in my opinion. One is synergism with antibiotics - this is a very important factor in treating bacteria-induced sepsis. The other one is the signaling properties of the oxygen level fluctuations - this partially explains why intermittent HBO2 sessions are more important than prolonging the single exposures (regardless of oxygen toxicity, correctly mentioned by the Authors).
Author Response
Thank you for taking the time to review our work. We are grateful for your insightful comments on our paper. We have incorporate changes to reflect the suggestions you provided. The changes are highlighted in the manuscript.
RE Synergism with antibiotics: We concur that the effects of HBO2 treatment on the invading pathogen are equally essential in bacteria-induced sepsis. This element has been addressed in the discussion section.
RE Signaling properties of oxygen level fluctuations: We agree that the activation of downstream signaling pathways in response to HBO2-mediated fluctuations in oxygen levels was not adequately described in the first manuscript; the revised manuscript incorporates this aspect in the introduction.
Reviewer 2 Report
Well-organized and comprehensive literature review and reasoned conclusion.
Language is clear and sophisticated. No need for statistical review. Appendix is comprehensive and useful/
Author Response
Dear Reviewer,
Thank you for taking the time to review our work. We appreciate your thoughtful remarks on our article. We have made a few minor modifications to the manuscript to further improve it. The modifications have been marked in the revised document.